# Heterogeneous Traffic Flow Signal Control and CAV Trajectory Optimization Based on Pre-Signal Lights and Dedicated CAV Lanes

**Jixiang Wang** [1,2], **Haiyang Yu** [1,3], **Siqi Chen** [1,4], **Zechang Ye** [1,2] and **Yilong Ren** [1,3,*]

1   School of Transportation Science and Engineering, Beihang University, Beijing 100191, China; jixiangwang@buaa.edu.cn (J.W.); hyyu@buaa.edu.cn (H.Y.); sy2213301@buaa.edu.cn (S.C.); ye_zc@buaa.edu.cn (Z.Y.)
2   Beihang Hangzhou Innovation Institute Yuhang, Hangzhou 310052, China
3   Zhongguancun Laboratory, Beijing 100083, China
4   Hefei Innovation Research Institute, Beihang University, Hefei 230071, China
*   Correspondence: serral@buaa.edu.cn

**Abstract:** This paper proposes a control system to address the efficiency and pollutant emissions of heterogeneous traffic flow composed of human-operated vehicles (HVs) and connected and automated vehicles (CAVs). Based on the comprehensive collection of information on the flow of heterogeneous traffic, the control system uses a two-layer optimization model for signal duration calculation and CAV trajectory planning. The upper model optimizes the phase duration in real time based on the actual total number and type of vehicles entering the control adjustment zone, while the lower model optimizes CAV lane-changing strategies and vehicle acceleration optimization curves based on the phase duration optimized by the upper model. The target function accounts for reducing fuel usage, carbon emission lane-changing costs, and vehicle travel delays. Based on the Webster optimal cycle formula, an improved cuckoo algorithm with strong search performance is created to solve the model. The numerical data confirmed the benefits of the suggested signal control and CAV trajectory optimization method based on pre-signal lights and dedicated CAV lanes for heterogeneous traffic flow. Intersection capacity was significantly enhanced, CAV average fuel consumption, carbon emission and lane-changing frequency were significantly reduced, and traffic flow speed and delay were significantly improved.

**Keywords:** pre-signal lights; CAV-dedicated lanes; heterogeneous traffic flow; signal control; trajectory optimization

## 1. Introduction

The development of Internet-connected autonomous vehicles has been accelerated by the quick advancement of both autonomous driving and Internet-connected technologies. The coexistence of CAVs and HVs on the road is an ineligible result [1,2]. Based on this, the transportation system emphasizes unique properties including networking, strong coupling, asymmetry, and uncertainty, substantially enhancing road traffic control intelligence while significantly escalating traffic management complexity [3–5].

Vehicle–road integrated collaborative decision-making technology is thought to be an effective way to handle heterogeneous traffic flow control and vehicle trajectory optimization in a vehicle–road collaborative environment. It is based on roadside observation devices to acquire vehicle operating information [6–8]. In order to operate the traffic flow in a safe, effective, and environmentally friendly manner, vehicle trajectories must be managed at the micro level and signal lights must be optimized at the macro level [9–12].

The current body of research on vehicle–road integrated collaborative optimization decision-making is primarily concerned with the connected vehicle scenario, and it may be

classified into two categories: pure CAV and CAV plus connected human-operated vehicle (CHV) vehicle–road integrated collaborative optimization decision-making technology. The former offers a higher level of vehicle intelligence, which can produce the best outcomes in lowering pollution emissions and minimizing traffic delays. It can even turn off signal lights automatically [13,14]. The latter focuses on how to increase overall traffic efficiency, coordinate the trajectories of cars approaching from different directions, and allow them to go through the conflict zone safely, effectively, and energy-efficiently [15].

Contrary to pure connected vehicle scenarios, where the problem of vehicle–road integrated collaborative optimization decision-making is present, the collaborative control of heterogeneous traffic flow at intersections involving CAVs and HVs necessitates taking into account the uncertainty of HVs' driving behavior because the accumulation of random factors (driver response time, speed tracking error, etc.) brought on by human drivers can increase the level of chaos in the entire transportation system [16,17]. As a result, the behavior of HVs can influence both the design of CAVs' trajectories and the optimization of signal lights. The most important difficulty is the safety danger brought on by HVs, which makes it very difficult to find a cooperative solution to the problem of controlling heterogeneous traffic flow in vehicle roads at intersections [18–20].

To lessen the uncertainty of a mixed traffic flow of CAVs and HVs, some researchers have first embraced two methods: CAV-dedicated lanes and HV trajectory prediction. To lessen the influence of HVs on the regulated system, the CAV-dedicated lane has been suggested. Talebpour [21] compared the three situations of full use, partial use, and prohibited use of CAV lanes and concluded that CAV lanes can effectively reduce road congestion and achieve energy conservation and emission reduction. According to Yu [22], research has shown that when CAVs have a high penetration, the system's efficiency is increased, while a low penetration results in a decrease in the system's safety. Right now, using CAV-dedicated lanes will dramatically decrease the frequency of risky circumstances while increasing the average speed. By merging a changeable virtual waiting area and a CAV-dedicated lane, Yu [23] created a three-layer optimization model that optimized barrier time, phase duration, waiting area changeover time, and CAV delay while preventing unpredictable HV driving behavior and cutting down on vehicle delay. Regarding HV trajectory prediction, B.-K. Xiong et al. [24] proposed a method based on random car-following model $\alpha$-percentile trajectories, where HVs have probabilities $\alpha$ and obey their $\alpha$-percentile trajectories. Guo [25] created a mixed traffic model based on cellular automata that forecasts the actions of human drivers based on data from the vehicles that are visible to them. It has been discovered that human drivers' attempts to learn more about the car in front of them have a major impact on the sub-stable state of traffic flow. A vehicle lane change choice model that takes efficiency and safety into account was put forth by Yu [26]. For the purpose of ensuring that AV decision execution is unaffected, it has been optimized based on the multi-player dynamic game model. To acquire driving intentions and optimize the trajectory of nearby vehicles, the lane-change trajectory is designed using polynomial curves and then verified through simulation using NGSIM (Next Generation Simulation).

Most of the above studies assumed that, on the one hand, the default HV also participates in the network connection, and on the other hand, the default HV does not exhibit lane-changing behavior. Both of these assumptions are unrealistic. If we take into account how an HV changes lanes when it is not connected to the network, it requires us to obtain the steering information of the HV upstream of the intersection in advance. Pre-signal lamps were initially used for bus priority [27]. In particular, this article innovatively collects HV lane change information by setting up a request for information zone with the assistance of pre-signal lights.

In this paper, our goals are to decrease vehicle fuel consumption and carbon emissions, and improve the efficiency of heterogeneous traffic flow at intersections. For isolated intersections, we divide the entrance lane into three parts: the request for information zone, the control adjustment zone, and the dedicated lane zone. We use pre-signal lights to gather in-

tersection phase information in the signal information area. In addition, with the assistance of pre-signal lights, we collect HV steering information through light language monitoring devices. We propose a methodology that utilizes pre-signal lights and dedicated lanes for CAVs to optimize signal control and vehicle trajectories, which includes optimization of signal phase duration, CAV trajectory optimization, and optimization of lane-changing loss, travel delay, energy consumption, and carbon emissions. The contributions of this work are as follows:

(1)  We introduce a new traffic management method based on pre-signal lights and CAV-dedicated lanes for signal management and CAV trajectory improvement of heterogeneous traffic movement, which integrates conventional signal timing strategies, dedicated CAV lanes, and CAV trajectory planning to manage the mixed traffic of CAVs and HVs. The method can be used in intersections where there are at least two lanes in each direction, which include key intersections commonly found in urban traffic control.

(2)  We propose dedicated CAV lanes that can be shared by straight-driving and left-turning CAVs. We perform CAV trajectory planning that includes lane-changing and intersections with HVs during the sharing phase to better utilize spatiotemporal resources and reduce vehicle fuel consumption and carbon emissions.

(3)  We used numerical modeling to calculate the energy consumption, carbon emissions, and delay of the vehicle under various situations. The experiments demonstrate that our proposed method significantly reduces the vehicle energy consumption and passage delay.

The remaining portions of this paper are arranged as follows: Section 2 describes the problem and provides the pertinent context. Section 3 introduces the coordinated model of signal and vehicle trajectories for heterogeneous traffic flow. Section 4 includes a discussion of the numerical studies. Section 5 provides a summary of the paper, along with recommendations for future research.

## 2. Problem Description and Background

Figure 1 illustrates a characteristic isolated intersection with a dedicated CAV lane, along with the layout of the road before the intersection. There are cars in each arm that can go in three different directions: left, straight forwards, and right. Left-turning and straight-driving CAVs are kept in the dedicated CAV lane. Both right-turning CAVs and right-turning HVs coexist in the right-turn lane and are not subject to signal regulation. The signal phase at intersections is split between HVs and CAVs. Each arm is organized into a request for information zone, a control adjustment zone, and a dedicated lane zone. In the request for information zone, pre-signal lights are installed, and the control system provides intersection phase information in advance with pre-signal lights. HVs provide steering information to the detection device through light language in the request for information zone under the pointer of the pre-signal lights and then go into the control adjustment zone. Due to the networking capabilities of CAVs, many sensing and communication tools are installed in the control adjustment zone to capture present-time information on the location, speed, acceleration, and lane selection of all vehicles in the area. CAVs complete trajectory planning, including lane-changing, based on the signal timing information obtained in advance in this area and then enter the dedicated lane zone. The dedicated lane zone is located between the adjusted control zone and the conflict zone and prohibits lane changing. After further optimizing the speed in the dedicated lane zone, vehicles will pass through the intersection without stopping. Based on the vehicle information collected in the request for information zone and control adjustment zone, the signal phase as well as CAV trajectory are dynamically adjusted to accomplish the objective of reducing energy expenditure and minimizing delay. To facilitate dynamic trajectory planning for CAVs and signal phase duration optimization, the intersection signal timing plan (phase duration), CAV lateral lane-changing strategy, and longitudinal speed adjustment are optimized in one framework. The coordinated optimization leverages the networking advantages of

CAVs, improves the quality of signal control, and minimizes energy consumption. For simplicity, in this paper, the following assumptions are made:

(1) The HV completes part of the lane change before entering the request for information zone and needs to complete at most one lane change operation in the control adjustment zone; HV drivers cooperate with the traffic rules in the request for information zone and provide light signal information without changing lanes.

(2) An HV's position, speed, acceleration, and other information can be detected by basic sensors and uploaded without delay.

(3) All CAV information in the control adjustment zone can be collected and uploaded without delay. In the control adjustment zone, CAVs can switch paths.

(4) Setting up the request for information zone and dedicated CAV lanes on the road and installing related sensing and communication equipment are allowed.

(5) The request for information zone is set up near a position with a distance of at least one traffic signal cycle length away from the intersection.

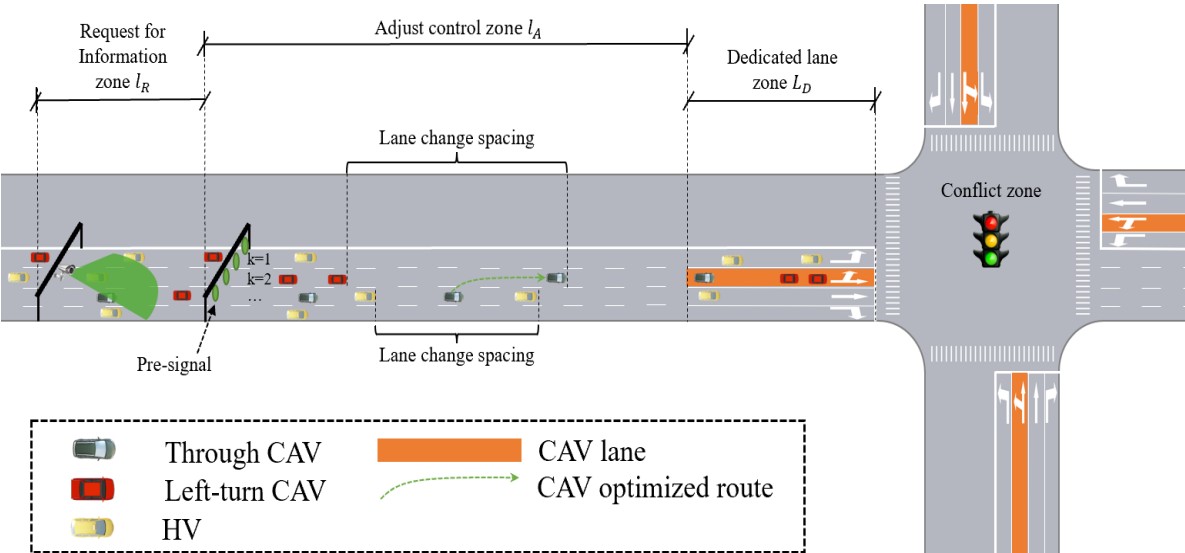

**Figure 1.** Typical isolated intersection approach arm layout.

## 3. Modelling and Solving

### 3.1. Model Framework

Figure 2 shows the duration of intersection signal lights $t_p$ and trajectory preparation of the CAV $\omega$ using a two-level optimization model framework. At the start time step $t_0^\omega$, the control system collects the semantic information of HV lights, CAV steering demand information, and velocity, as well as position information of each vehicle in the control adjustment zone and dedicated lanes.

During the initialization phase, given that the direction of movement of the HV within the adjusted control zone has been obtained in the request for information zone, due to the deployment of a multitude of sensors in the control adjustment zone, the position and speed information of the HVs can be obtained in a timely manner. Therefore, the trajectory of $V_H$ in front of CAV $\omega$ can be predicted using the second-order vehicle-following model proposed by Eissfeldt [28] and the improved lane change model established by Erdmann [29]. Considering that there are a plethora of sensor devices deployed in the control adjustment zone, it is possible to calculate the total amount and kind of vehicles in the control adjustment zone and the dedicated lane zone, which has a direct impact on the phase maintenance time. Since a dedicated lane is set for CAV in the dedicated lane zone, the CAV $\omega$ directly receives phase duration information. The CAV in front of CAV $\omega$ will

follow the planned trajectory. Therefore, for CAV $\omega$, due to the tracking information of the vehicle in front, CAV $\omega$ will generate a feasible trajectory.

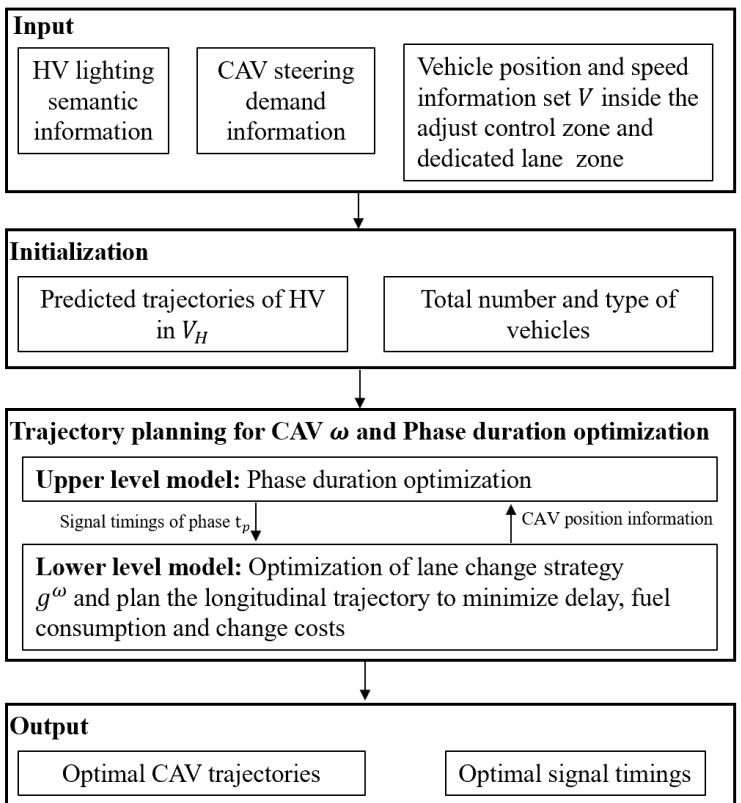

**Figure 2.** Model framework.

Regarding the two-layer model for signal phase duration optimization and trajectory planning of CAV $\omega$, the phase maintenance time $t_p$, the CAV lane change strategy $g^\omega$, and the curve of longitudinal acceleration $a^\omega$ are cooperatively optimized. In the model of the top layer, the phase duration $t_p$ can be optimized on the spot according to the total number and type of actual vehicles entering the request for information zone, and in the lower-layer model, the lane-change method information $g^\omega$ and the vehicle acceleration optimization curve $a^\omega$ are optimized to achieve the maximum reduction of delay time, fuel consumption and lane change cost of CAV $\omega$. At the same time, the lower layer also sends the position data of CAV $\omega$ to the upper-layer model in time to avoid phase waste. The two layers of models influence each other.

The output of the two-layer model is the intersection length of the signal phase $t_p$ optimization as well as the optimal trajectory planning of CAV $\omega$. In trajectory planning, this paper constrains the behavior of CAV $\omega$ to have no effect on the behavior of other vehicles, and the signal phase optimization is constrained by the intersection signal minimum and maximum durations.

### 3.2. Upper-Layer Model: Signal Light Phase Optimization

Given the signal timing (straight ahead then left turn) and the total number and type of vehicles entering the request for information zone, this paper uses Webster's optimal cycle formula [30] to calculate the cycle duration and then uses the saturation ratio to assign the best green light time to every phase, as shown in Formulas (1)–(5),

$$C_{opt} = \frac{1.5L + 5.0}{1.0 - Y} \tag{1}$$

$$G_e = C_{opt} - L \tag{2}$$

$$L = n \times (l + r_{e,i}) \tag{3}$$

$$g_{e,i} = G_e \frac{y_i}{Y} \tag{4}$$

$$t_p = g_{e,i} \tag{5}$$

where $C_{opt}$ is the optimal signal period, $L$ is the total time lost at intersections, $Y$ represents the total of the intersection traffic flow ratios, $G_e$ is the effective green light time, $n$ is the number of phases of the signal, $l$ is the loss time of the signal, $r_{e,i}$ is the red light time of phase $i$, $y_i$ is the traffic flow ratio of the critical lanes of every phase, and $g_{e,i}$ is the effective green light time of phase $i$.

The relevant constraint of Equation (4) is

$$g_{min} \leq g_{e,i} \leq g_{max} \tag{6}$$

where $g_{min}$ is the shortest green light period in the phase and $g_{max}$ is the longest green light period in the phase.

### 3.3. Lower-Layer Model: Optimized Lane Change Strategy

The model of the lower layer, formed using the phase maintenance time $t_p$ offered by the upper layer, achieves the objective of minimizing the CAV $\omega$ passage time delay, reducing the CAV $\omega$ fuel economy as well as the cost of lane change by selecting a suitable lane change strategy $g^{\omega}$ and formulating a reasonable longitudinal acceleration optimization curve $a^{\omega}$ for CAV $\omega$. These goals can be summarized by Equation (7) as follows:

$$\min_{g^{\omega}} C[a^{\omega}, g^{\omega}] = \theta_1 \sum_{t=t_0^{\omega}}^{t_0^{\omega}+h_1+h_2} \delta^{\omega}(t)\Delta t + \theta_2 \sum_{t=t_0^{\omega}}^{t_0^{\omega}+h_1+h_2} \delta^{\omega}(t)|a^{\omega}(t)| + \theta_3 \sum_{t=t_0^{\omega}}^{t_0^{\omega}+h_1} \vartheta^{\omega}(t) \tag{7}$$

where $t_0^{\omega} + h_1 + h_2$ is the optimized duration of CAV $\omega$ between the request for information zone and the conflict zone, $t_0^{\omega}$ is the moment when CAV $\omega$ begins trajectory optimization in the request for information zone, $h_1$ is the optimization duration of CAV $\omega$ in the control adjustment zone, and $h_2$ is the optimization duration of CAV $\omega$ in the dedicated lane zone. If CAV $\omega$ drives into the conflict zone, $\delta^{\omega}$ is 0; otherwise, it is 1. In the request for information zone, if CAV $\omega$ has no lane-changing behavior, $\vartheta^{\omega}$ is 0; if there is lane-changing behavior, $\vartheta^{\omega}$ is 1. Vehicle delay usually refers to the difference between the real passage time and free exercise time, so narrowing the gap between the two is one of the objectives of concern for model control. The smoothing degree of vehicle acceleration also has an essential effect on vehicle emissions. The proportion of lane changes of CAV $\omega$ has a meaningful effect on the cost of lane changes, and since this article focuses on the problem of CAV passage efficiency and energy consumption reduction, the weight coefficients of the different control objectives $\theta_1$, $\theta_2$ and $\theta_3$ are not the same, that is, $\theta_1 > \theta_2 \gg \theta_3$.

When formulating the lane change strategy $g^{\omega}$ for CAV $\omega$, the order of lane change actions must be constrained. Equations (8) and (9) describe the constraints related to the sequence of lane change actions for CAV $\omega$.

$$-M(2 - \vartheta^{\omega}(t_1) - \vartheta^{\omega}(t_2)) + \epsilon_{lc} \leq (t_2 - t_1)\Delta t, \\ \forall t_2 = t_1 + 1, \cdots, t_0^{\omega} + h_1; \; t_1 = t_0^{\omega} + 1, \cdots, t_0^{\omega} + h_1; \tag{8}$$

$$|k(g^{\omega}(t+1)) - k(g^{\omega}(t))| = \vartheta^{\omega}(t+1), \\ \forall t = t_0^{\omega} + 1, \cdots, t_0^{\omega} + h_1 - 1 \tag{9}$$

where $\epsilon_{lc}$ is the lowest time interval between two consistent lane change operations and $k(g^{\omega}(t))$ is the lane number that CAV $\omega$ is in at moment $t$. Equation (8) constrains the duration between two consecutive lane changes of the CAV $\omega$, requiring it to be greater than the minimum time interval. Equation (9) constrains the CAV $\omega$ to change lanes at most one lane at a time.

In addition to satisfying the above constraints, the vehicle must also satisfy the vehicle kinematic constraints and longitudinal safety distance constraints.

Given the position $x^{\omega}(t_0^{\omega})$, velocity $v^{\omega}(t_0^{\omega})$, acceleration $a^{\omega}(t_0^{\omega})$, and lane $k^{\omega}(t_0^{\omega})$ of CAV $\omega$ at time $t_0$, before $\omega$ enters the conflict zone of the intersection, the vehicle's motion follows a second-order vehicle kinematic model:

$$v^{\omega}(t+1) = v^{\omega}(t) + a^{\omega} \times \Delta t,$$
$$\forall t = t_0^{\omega}, t_0^{\omega} + 1, \cdots, t_0^{\omega} + h_1 + h_2 - 1 \tag{10}$$

$$x^{\omega}(t+1) = x^{\omega}(t) + \frac{\Delta t}{2} \times (v^{\omega}(t) + v^{\omega}(t+1)),$$
$$\forall t = t_0^{\omega}, t_0^{\omega} + 1, \cdots, t_0^{\omega} + h_1 + h_2 - 1 \tag{11}$$

$$-a_L^{\omega} \leq a^{\omega}(t) \leq a_U^{\omega},$$
$$\forall t = t_0^{\omega}, t_0^{\omega} + 1, \cdots, t_0^{\omega} + h_1 + h_2 \tag{12}$$

$$0 \leq v^{\omega}(t) \leq v_U^{\omega},$$
$$\forall t = t_0^{\omega}, t_0^{\omega} + 1, \cdots, t_0^{\omega} + h_1 + h_2 \tag{13}$$

where $a_U^{\omega}$ and $a_L^{\omega}$ are the highest acceleration and highest deceleration of the vehicle, respectively, and $v_U^{\omega}$ is the vehicle speed limit on the road section.

To establish the reliable operation of the vehicle, the vehicle should maintain a safe longitudinal distance with the front and rear vehicles, and the safe longitudinal distance is not the same for all speeds. For the convenience of calculation, this paper selects the Newell vehicle-following model to determine the following constraints.

$$x^{\omega}(t) \leq x^{\omega'}\left(t - \frac{\tau_{cf}}{\Delta t}\right) - d_{cf}$$
$$\forall \omega' = V_p(g^{\omega}(t)); t = t_0^{\omega} + \frac{\tau_{cf}}{\Delta t}, t_0^{\omega} + \frac{\tau_{cf}}{\Delta t} + 1, \cdots, t_0^{\omega} + h_1 + h_2 \tag{14}$$

$$x^{\omega}(t) \leq l_A + l_D \tag{15}$$

$$x^{\omega}(t) \geq x^{\omega'}\left(t + \frac{\tau_{cf}}{\Delta t}\right) + d_{cf},$$
$$\forall \omega' = V_f(g^{\omega}(t)) \in \Omega^{\omega}; t = t_0^{\omega}, t_0^{\omega} + +1, \cdots, t_0^{\omega} + h_1 + h_2 - \frac{\tau_{cf}}{\Delta t} \tag{16}$$

$$x^{\omega}(t) \geq x^{\omega'}(t) + \frac{(v^{\omega'}(t))^2}{2a_L^{\omega'}} - M(1 - \vartheta^{\omega}(t)),$$
$$\forall \omega' = V_f(g^{\omega}(t)) \in \overline{\Omega}^{\omega}; t = t_0^{\omega}, t_0^{\omega} + 1, \cdots, t_0^{\omega} + h_1 + h_2 \tag{17}$$

$$x^{\omega}(t) \leq l_A + M(1 - \vartheta^{\omega}(t)) \tag{18}$$

where $V_p(g^{\omega}(t))$ is the set of leading vehicles ahead of CAV $\omega$, and $V_f(g^{\omega}(t))$ is the set of following vehicles behind CAV $\omega$. Equation (14) is the distance constraint between vehicle $\omega$ ahead in the same lane and CAV $\omega$. $\tau_{cf}$ and $d_{cf}$ are the temporal displacement and spatial displacement in the Newell vehicle-following model, respectively, and $\Delta t$ should be reasonably set to make $\frac{\tau_{cf}}{\Delta t}$ an integer. Similarly, Equation (16) is the distance constraint between the rear following vehicle $\omega'$ and CAV $\omega$ in the same lane.

### 3.4. Model-Solving Algorithms

In the two-layer model, the signal light phase optimization is relatively simple, while the lane change strategy $g^{\omega}(t)$ and acceleration $a^{\omega}(t)$ optimization in the lower model is a complex nonlinear optimization problem with multiple constraints and multiple variables.

Thus, there is a problem of a large solution space. For this reason, we design an improved cuckoo algorithm solution model with strong search performance combined with the EO algorithm based on the optimal phase duration calculated by using Webster's optimal cycle formula. We find the optimal solution for the objective function established in Section 3.3.

The cuckoo search algorithm is an algorithm for a flight search mechanism inspired by the biological phenomenon of cuckoos parasitically hatching their young, and nest locations are updated in the following manner.

$$X_i^{t+1} = X_i^t + a \oplus Levy(\lambda) \tag{19}$$

$$\alpha = \alpha_0 \left( X_i^t - X_{best} \right) \tag{20}$$

$$Levy(\lambda) = \frac{u}{|v|} \tag{21}$$

$$u \frown N\left(0, \delta_u^2\right) \tag{22}$$

$$v \frown N(0,1) \tag{23}$$

$$\delta_u = \left\{ \frac{\Gamma(\lambda)\sin(\pi(\lambda-1)/2)}{\Gamma[(\lambda)/2](\lambda-1)2^{(\lambda-2)/2}} \right\}^{1/(\lambda-1)} \tag{24}$$

where $X_i^t = (x_{i1}, x_{i2}, \cdots, x_{in})(i = 1, 2, \cdots, m)$ denotes the position of the $i$th nest in the $t$th generation; $n$ is the dimension of the optimization problem; $m$ is the number of nests per generation; $\alpha$ is the step size factor, as a control of the random search range; $\alpha_0$ is a constant, often taken as 0.01; $X_{best}$ is the current optimal solution; the symbol $\oplus$ denotes elementwise multiplication; and $Levy(\lambda)$ is the random search vector generated by the Rheinland flight obeying the parameter $\lambda$ ($1 < \lambda \leq 3$).

Since the cuckoo algorithm tends to fall into local extrema, it has the problem of premature aging. In this paper, the polar dynamics optimization algorithm is established in the framework of the cuckoo algorithm to improve the speed and correctness of model solving. For a polarization problem, the flow of the EO algorithm is as follows.

(1) For an individual $X_i^t = (x_{i1}, x_{i2}, \cdots, x_{in})$, let the optimal solution searched thus far be $X_{best}$ and its objective function value be $C(X_{best})$.

(2) Perform the following operations on the current individual $X_i^t$:

    (a) Calculate the fitness $\lambda_{ij}$ of each group element $x_{ij}$, $j \in \{1, 2, \cdots, n\}$;

    (b) Sort the $n$ fitnesses to find the group element $x_{min}$ with the smallest fitness: $\lambda_{min} \leq \lambda_{ij}$, $j = 1, 2, \cdots, n$; then $x_{min}$ is the worst group element;

    (c) Find a neighbour $X_i^{t*}$ in the neighbourhood of the current individual $X_i^t$ to force the worst group element $x_{min}$ to change;

    (d) Unconditionally accept $X_i^t = X_i^{t*}$;

    (e) If the current objective function value is $C(X_{best})$,

    then $X_{best} = X_i^t$, $C(X_{best}) = C(X_i^t)$.

(3) Repeat step (2) until the termination condition is satisfied.

(4) Obtain the solution $X_{best}$ and the optimal value of the cost function $C(X_{best})$.

Combining the Webster optimal cycle formula and the improved cuckoo algorithm, the model-specific solution process is obtained as follows, and the method flow chart is illustrated by Figure 3.

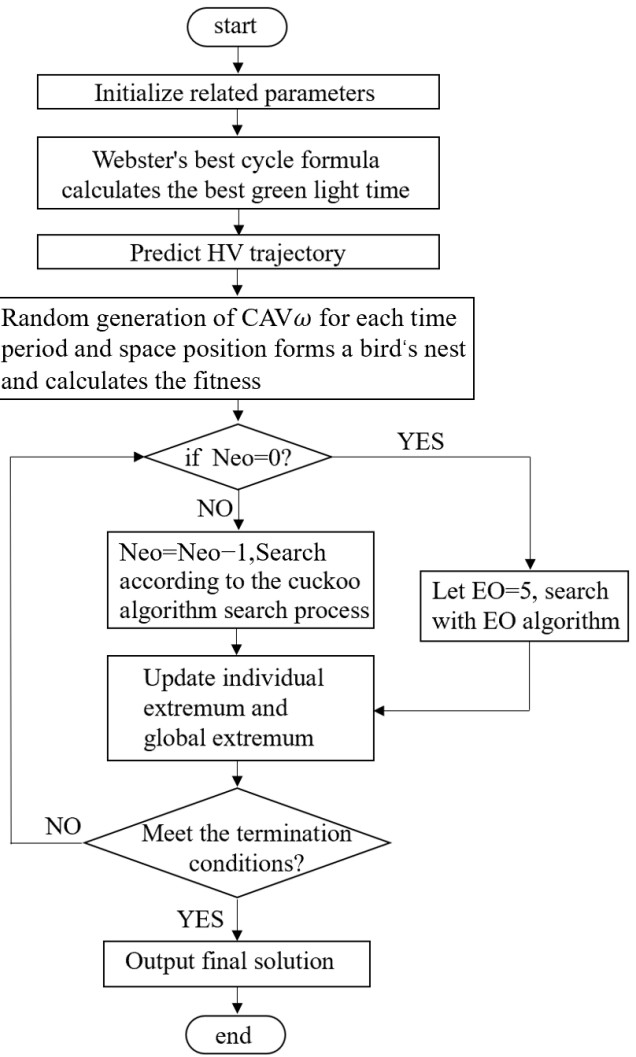

**Figure 3.** Algorithm flow chart.

Step 1: Set the parameters related to the algorithm and let Neo = 5. Neo refers to the interval algebra in the cuckoo algorithm. The signal control system calculates the cycle time based on the given signal timing and the whole number and type of vehicles entering the request for information zone using the Webster optimal cycle formula and assigns the optimal green light time $t_p$ to each phase using the saturation ratio.

Step 2: Starting from time $t_0^\omega$, CAV $\omega$ obtains the operation status of vehicles in the request for information zone, the phase maintenance time $t_p$ information sent down from the upper model, and the trajectory planning information of the CAV ahead.

Step 3: Combining the semantic information of HV lane change provided by the pre-signal, the future trajectory of HVs in the request for information zone and the dedicated lane zone is predicted based on the car-following model of Eissfeldt and the improved lane change model of Erdmann to plan the current forward trajectory of CAV $\omega$.

Step 4: Based on the planned trajectory and predicted trajectory information of the vehicle ahead, combined with the relevant parameters of the cuckoo algorithm, a bird's nest is randomly generated for each period CAV $\omega$ spatial location composed of a bird's nest. Each bird's nest $X_i^t$ has a dimension of $N \times T$, where $N$ is the number of controlled variables. The adaptation degree of each nest is calculated, and the optimal position nest $X_{best}$ is solved.

Step 5: Determine if Neo is 0. If Neo is 0, go to step 7; otherwise, continue to the next step.

Step 6: Generate a new position for each nest according to the Levy flight and calculate its fitness; go to step 8.

Step 7: An optimization search is performed for each nest based on EO search, and the fitness of each nest is calculated.

Step 8: Update the individual minimum and global minimum: for each nest, calculate its fitness and compare it with the individual value; if it is better, replace the current individual extreme value and the individual optimal position. Afterwards, compare the fitness of each nest with the global optimal; if it is better than the global optimal, update the current global extreme value and the position of the global optimal nest.

Step 9: Determine whether the ultimate number of recurrences has been achieved, and if not, proceed to step 5 to continue the execution; otherwise, proceed to the next step.

Step 10: Output the optimal phase maintenance time $t_p$ and the optimal trajectory for CAV $\omega$.

## 4. Simulation Analysis

### 4.1. Numerical Settings

To analyze the performance of the optimization strategy offered in this paper, a simulation scenario as in Figure 1 is set up, with four arms at the intersection, each arm having four lanes containing HV left-turn lanes, CAV-dedicated lanes, HV straight lanes and right-turn-only lanes, with no delay in communication. The signal control scheme is for the vehicles to go straight ahead first and then turn left. The length of the request for information zone is 100 m, the length of the control adjustment zone is 500 m, and the length of the dedicated lane zone is 30 m. The velocity limit of the request for information zone, control adjustment zone, and dedicated lane zones is $v_U^\omega$ 13.8 m/s, and the velocity limit of the conflict zone is 8.3 m/s.

To better demonstrate the advantages of signal phase enhancement and CAV trajectory generation and to eliminate the differences brought by driving behaviors, CAVs and HVs use the same parameters in the driving model settings, and the body length is 4 m. The spatial and temporal displacements in the car-following model are 6 m and 1 s, respectively. The minimal time interval $\epsilon_{lc}$ between successive lane change behaviors is 5 s, and the maximum vehicle acceleration $a_U^\omega$ and minimum acceleration $a_L^\omega$ are 2 m/s² and 4 m/s², respectively. $d_p$ and $d_f$ are set to 5 m and 6 m, respectively, in a conservative manner in lane change planning. The weighted parameters $\theta_1$, $\theta_2$, $\theta_3$ in the objective function of the lower-layer model are 1000 s⁻¹, 10 m/s² and 1, and the time step $\Delta t$ is set to 1 s. The vehicle entry conforms to a Poisson distribution, and the starting speed of the CAV vehicle entering the regulation area is generated randomly between $v_0^L$ (set to 3 m/s in this paper) and $v_0^U$ (set to 13.8 m/s in this paper). The driving behavior of the HV in the simulation was captured using Wiedermann 74 in VISSIM 4.3.

The average vehicle arrival rate for each arm is 4200 veh/h. The vehicle arrival during the simulation contains different traffic demands from undersaturated to oversaturated, where the divisions of vehicles going straight, turning left, and turning right are 0.4, 0.4 and 0.2, respectively, and the penetration of CAV is 50%, except for during the CAV penetration analysis. In the upper model, the shortest green light time $g_{e,i}^{min}$ = 12 s, the maximal green light time $g_{e,i}^{max}$ = 18 s, and the spacing time $r_{e,i}$ = 2 s are set. In this paper, the simulation model is written in Python 3.7.4, and all experimental procedures are executed on a desktop computer with 16 GB of memory and an Intel 2.6 GHz CPU. To improve computational efficiency, five threads are employed in parallel. During each experiment's simulation, five random seeds are employed while factoring into the uncertainty surrounding vehicle arrival and HV driving behavior. Each simulation is run for 1200 s with a warm-up time of 150 s.

### 4.2. Analysis of Results

4.2.1. Simulation Results and Analysis

Given that this paper addresses the problem of cooperative perfect control of CAV path planning as well as signalization for traffic flow that is diverse at a particular intersection, intersection maximization capacity along with CAV minimization delay and fuel economy are used as the evaluation indices to evaluate and analyze the performance merits of the signal management and CAV trajectory models in heterogeneous traffic flow based on the method proposed in this paper.

Since this paper sets up the request for information zone and pre-signal for vehicles before the CAV enters the control adjustment zone, the upper-layer optimal control model can request the type and forwards direction of vehicles that are about to enter the request for information zone and the dedicated lane zone in time. This aids intersection signal optimization and capacity improvement. It should be emphasized that there are few suitable benchmark studies in the literature that can be used as a fair comparison, as existing studies rarely consider the lane-changing behavior of heterogeneous traffic flows. To better analyze the merits of the control model proposed in this research, the VISSIM fixed timing signal control model without the request for information zone is chosen as a control. The cycle length of the signal plan used in the fixed timing signal model is 60 s. The green period of both the straight phase and the left-turn phase is 14 s, and the yellow duration is 2 s. The signal does not regulate the right-turning vehicles. Figure 4 shows the results of simulating the vehicle throughput conditions, delay conditions, average vehicle speed, and average number of lane changes for heterogeneous traffic flows under two different signal control strategies.

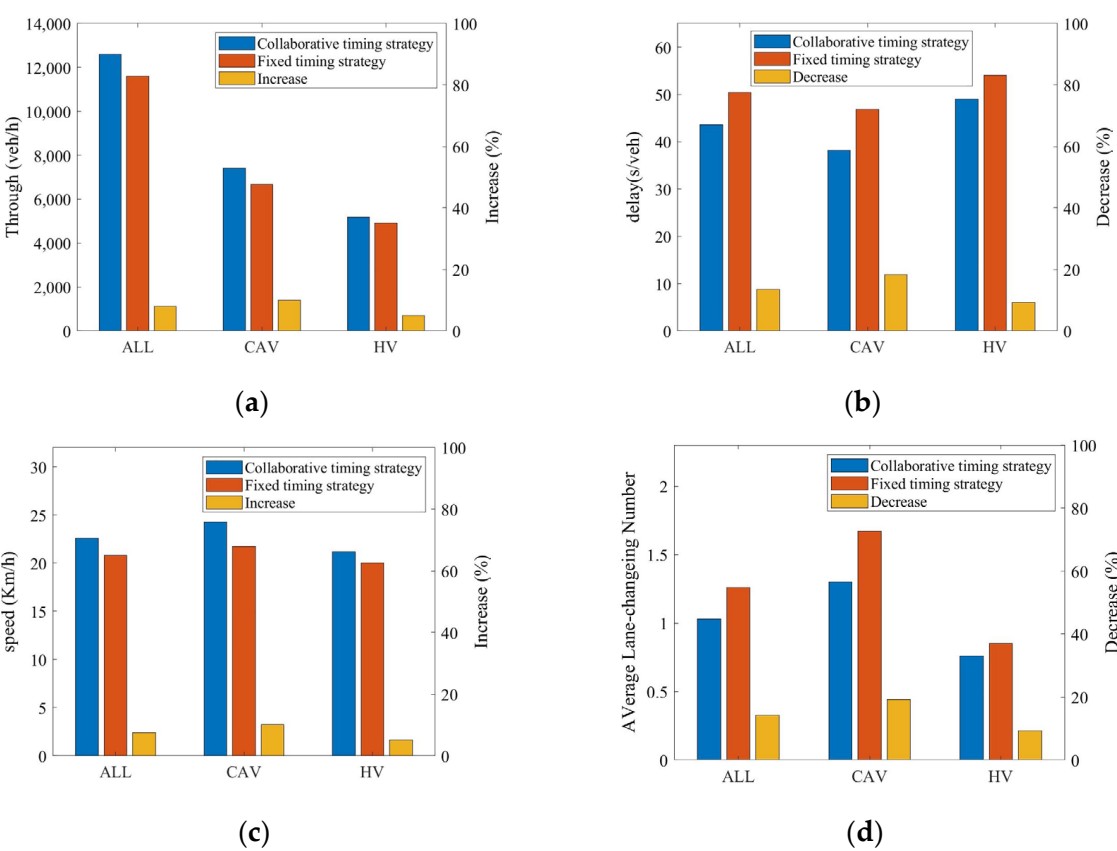

**Figure 4.** Comparison of the effects of two control strategies. (**a**) Vehicle throughput, (**b**) Vehicle delay, (**c**) Average vehicle speed, (**d**) Average number of lane changes.

From Figure 4a, when cars appear at the intersection with a Poisson distribution, the cooperative timing strategy proposed in this research significantly improves the intersection

resources compared with the fixed timing strategy, increasing the efficiency of all vehicles, CAVs, and HVs by 8%, 10%, and 5.1%, respectively. In comparison, the proposed request for information zones and pre-signals help to increase the capacity of the intersection and have a larger effect on increasing the capacity of CAVs. The primary cause for this result is that the planned request for the information zone allows intersections to obtain HV turn information in advance, which helps CAV trajectory planning and signal control and distributes the signal control time reasonably. In addition to improving the throughput, from Figure 4b, when cars reach the intersection with a Poisson distribution, the cooperative timing strategy proposed in this paper significantly improves the throughput delay of heterogeneous traffic flow at the intersection compared with the fixed timing strategy. The throughput delay of all vehicles, CAVs, and HVs is reduced by 13.5%, 18.4%, and 9.26%, respectively. This further confirms the effectiveness of the proposed request for information zones and pre-signals to reduce vehicle delays. The reason for this change is that the advanced acquisition of HV turn information by the request for information zones helps the control system make signal control adjustments, reduces conflicts between heterogeneous traffic flows due to lane changes, and increases the average speed of vehicular traffic. This assertion can also be confirmed by the statistics of the average velocity and average number of lane changes of vehicles in heterogeneous traffic flow from the adjustment zone to the conflict zone, as shown in Figure 4c,d. Compared with the fixed timing strategy, the cooperative timing strategy considered in this article improves the average speed of all vehicles, CAVs and HVs in the heterogeneous traffic flow at the intersection by 8.54%, 11.6%, and 5.9%, respectively, and decreases the average number of lane changes by 16.38%, 22.15%, and 10.6%, respectively.

In terms of vehicle fuel economy and $CO_2$ emissions, it is plain to see from Figure 5 that when cars arrive at the intersection with a Poisson distribution, the cooperative timing strategy considered in this article improves the fuel economy of heterogeneous traffic flows compared with the fixed timing strategy, improving the fuel economy of all vehicles, CAVs, and HVs by 12.9%, 17.9%, and 8.8%, respectively, and the $CO_2$ emissions of all vehicles, CAVs, and HVs by 13.5%, 19.1%, and 9.6%, respectively. The vehicle fuel consumption and $CO_2$ emissions are estimated using the fuel consumption model [31]. The spatiotemporal trajectory of the CAV in Figure 6a shows that due to the advanced acquisition of intersection signal control information and HV turning information, the CAV can follow a more reasonable trajectory to avoid vehicle stalling, which in turn increases the capacity of the intersection, improves vehicle speed, and makes full use of green light time. To prove this point in depth, a random CAV is selected to analyze the acceleration change under the two different control strategies, as shown in Figure 6b. There is a clear distinction between the fixed timing strategy and the cooperative timing strategy proposed in this article in terms of reducing the speed fluctuations of the CAV.

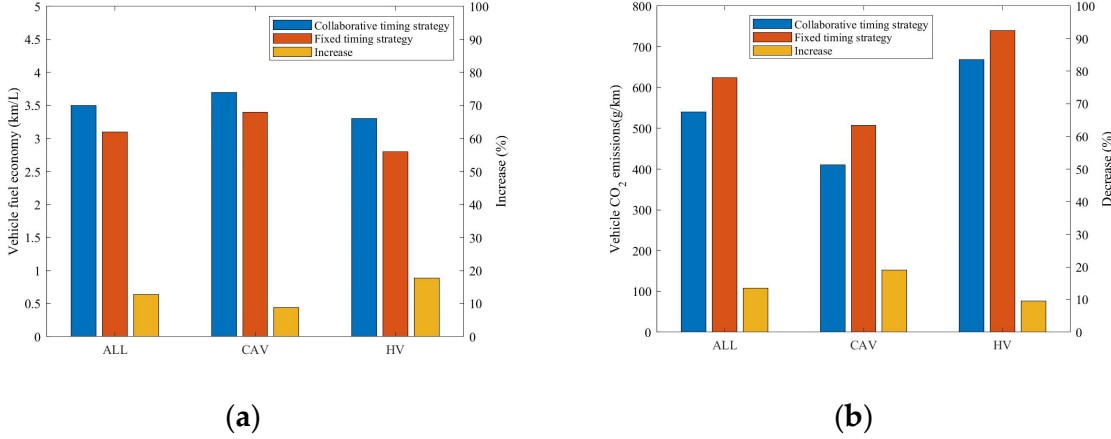

**Figure 5.** Fuel economy and $CO_2$ emissions. (**a**) Fuel economy. (**b**) $CO_2$ emissions.

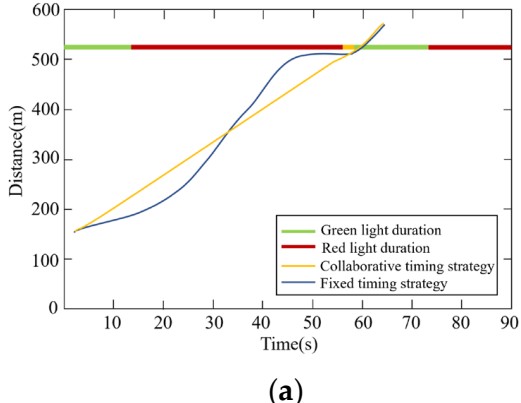
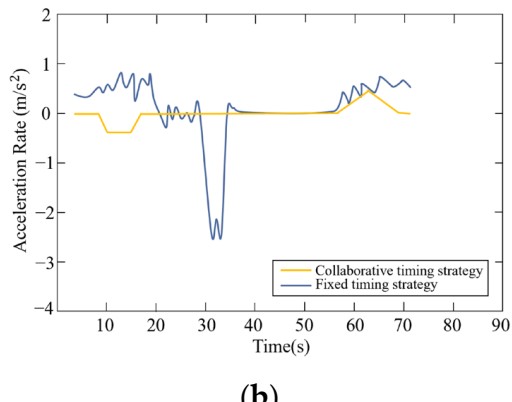

**(a)**

**(b)**

**Figure 6.** Trajectory and acceleration conditions. (**a**) Space-time trajectory of a CAV. (**b**) Acceleration profiles.

4.2.2. Sensitivity Analysis

As analyzed previously, the optimization control model proposed in this article has a positive impact on the passage of heterogeneous traffic flow at intersections, helping to reduce the delay of heterogeneous traffic flow, decrease $CO_2$ emissions, and improve average fuel economy while increasing the intersection's traffic capacity. To further analyze the benefits of the recommended model, this section explores the impact of the control model on heterogeneous traffic flow under different traffic demands as well as the impact of various CAV penetration rates.

Figure 7a–c display the average delay for heterogeneous traffic flow, CAVs, and HVs when the saturation flow rates are 0.5, 1.0, and 1.25, respectively. As traffic demand increases, the typical delays for mixed traffic flow, CAVs, and HVs all increase significantly. When the CAV penetration rate rises, the average delay for mixed traffic flow and HVs decreases significantly for each demand level. This indicates that signal control for heterogeneous traffic flow based on pre-signal lights and CAV-dedicated lanes, as well as CAV trajectory optimization, can reduce the delay of mixed traffic flow and HVs when CAVs are more prevalent in mixed traffic flow, and the reduction in delay is most significant for HVs. Figure 7d–f show the reduction in general delay of heterogeneous traffic flow, CAVs, and HVs under different demand levels in contrast to the baseline case without CAV trajectory design and signal coordination optimization. Although CAVs' trajectories are planned in a dispersed manner, their travel delay is significantly improved. When the rate of CAV penetration is high (70%), the reduction in delay for HVs also becomes significant, especially when the flow rate is excessively saturated (V/C = 1.25), with a delay reduction of over 7 s. The reason is that under a more reasonable phase timing environment, the proposed pre-signal lights and CAV-dedicated lanes provide more time and space resources for HV travel, allowing HVs to make adjustments in advance to pass through intersections at higher speeds, avoiding emergency braking and acceleration of vehicles and avoiding stopping at the stop bar. In Figure 8, the trajectories of HVs under different control strategies are illustrated using lane 3 (straight-through lane) as an example when 70% of CAVs are inserted. The trajectory segments in differing lanes are evident in light colours. In Figure 8a, the HV stops at the red light and starts when the light is green, resulting in significant fluctuations in the HV trajectories. Unfortunately, these HVs lose start-up time. Conversely, in Figure 8b, the HVs can pass smoothly and quickly through the stop bar, avoiding stopping at the stop bar and effectively utilizing the green light time. Pre-signal lights and CAV-dedicated lanes can effectively reduce the delay of HVs.

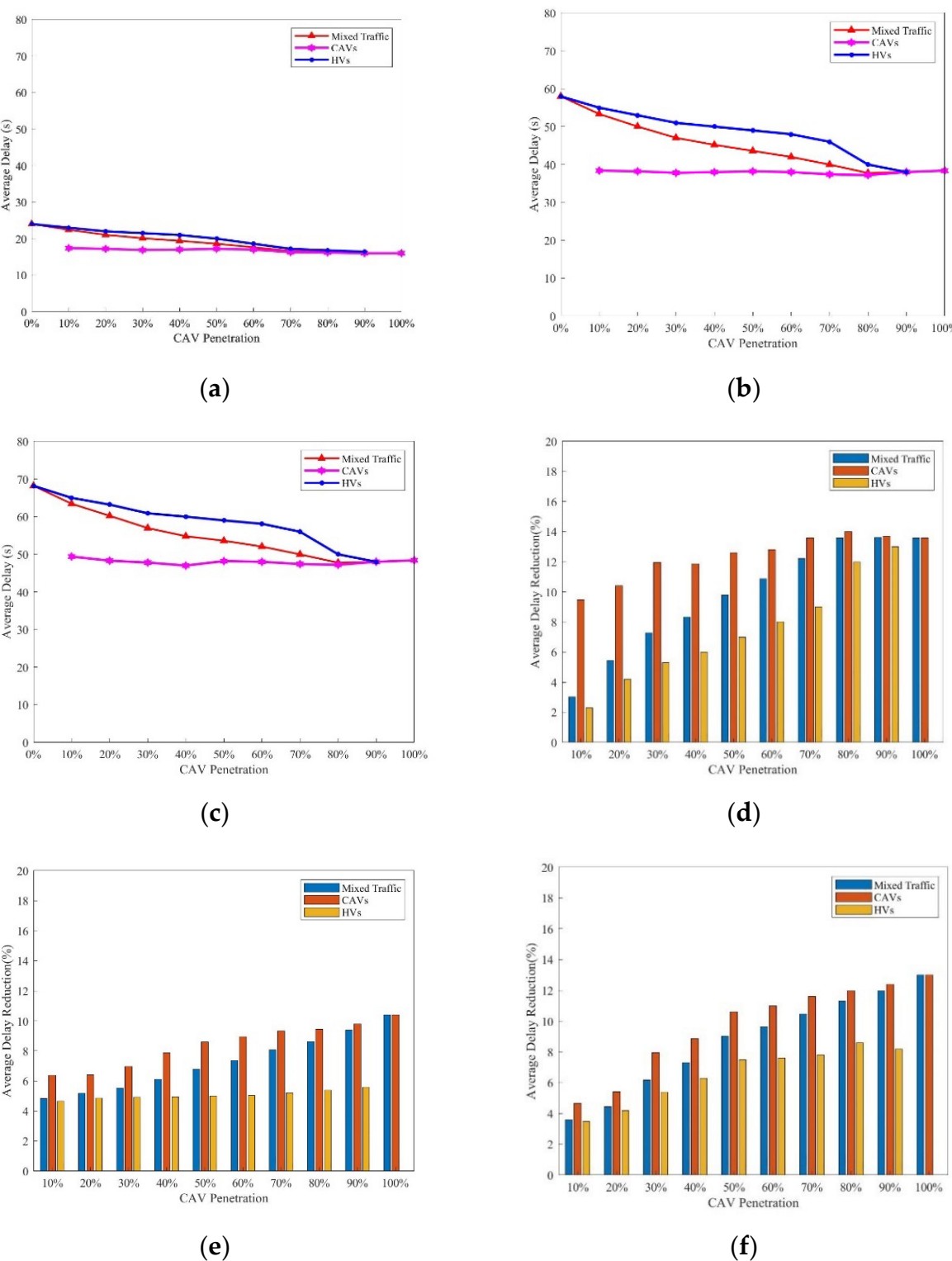

**Figure 7.** Impact of CAV penetration rate on vehicle delay under different traffic demands. (**a**) Average delay (V/C = 0.50); (**b**) Average delay (V/C = 1.00); (**c**) Average delay (V/C = 1.25); (**d**) Average delay reduction (V/C = 0.50); (**e**) Average delay reduction (V/C = 1.00); (**f**) Average delay reduction (V/C = 1.25).

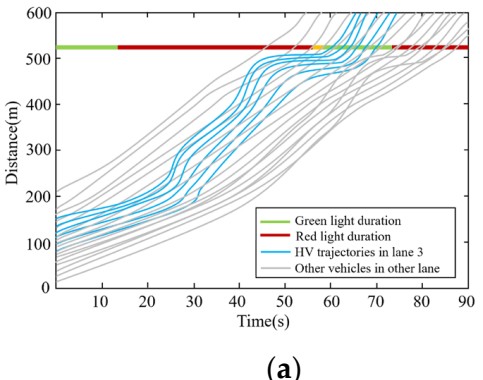 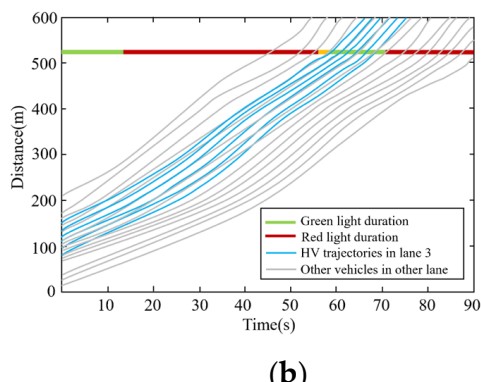

**(a)** **(b)**

**Figure 8.** Comparison of HV trajectories under different strategies. (**a**) HV trajectory under the fixed timing strategy; (**b**) HV trajectory under the pre-signal light and CAV-dedicated lane model.

In addition, for similar reasons, the average passing capacity of intersections is also improved. Figure 9 compares and analyzes the positive effects of the CAV penetration rate on intersection passing capacity under different traffic demands. The tested CAV percentages were 0%, 25%, 50%, 75%, 85%, and 100%. The passing capacity reaches its upper limit when the traffic volume line becomes flat. Figure 9 shows that when the prevalence of CAV rises from 0% to 50%, the passing capacity is significantly improved, reaching 9.16%. When all vehicles are CAVs, a significant enhancement is observed, and the passing capacity reaches 13.36%. An interesting phenomenon is that the maximum enhancement is achieved when the CAV penetration rate is 85%, reaching 15.27%. Because of CAV-dedicated lanes, too low a percentage of HVs means that the traffic resources of HV lanes are idle. Therefore, the best effect is achieved when the CAV percentage is 85%, rather than 100%. These observations indicate that CAV trajectory organization can significantly enhance the passing capacity of intersections, but it requires a higher CAV penetration rate, and the setting of CAV-dedicated lanes has a significant influence on improving the passing capacity of heterogeneous traffic flow.

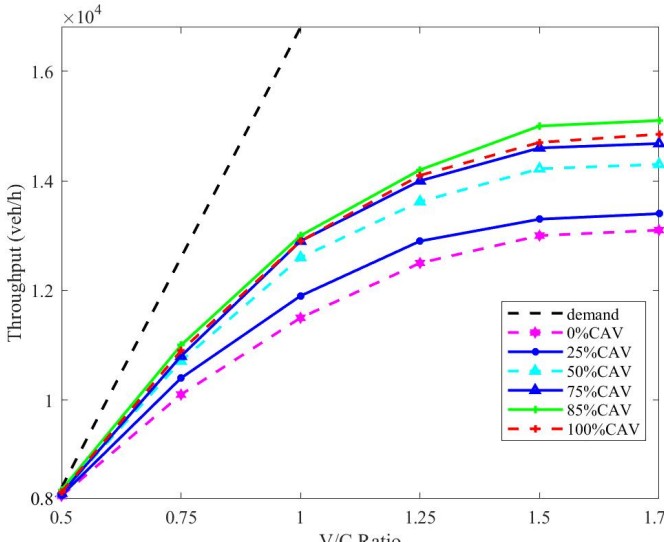

**Figure 9.** Impact of CAV penetration rate on intersection passing capacity under different demands.

Figure 10a,c,e depict the average fuel economy of a variety of transportation flows. Figure 10b,d,f depict the average $CO_2$ emission of a variety of transportation flows. The traffic demand conditions for CAVs and HVs are different for different CAV penetration rates. As traffic demand rises, the average fuel economy of heterogeneous traffic flow,

CAVs, and HVs significantly decreases. The increased rate of CAV penetration leads to a significant improvement in fuel economy, especially under high traffic volumes. There are two reasons for this: on the one hand, appropriate signal control strategies provide reasonable travel time for heterogeneous traffic flow; on the other hand, CAV trajectory organization based on dedicated CAV lanes helps to avoid the stop-and-go motion of mixed traffic and smooth longitudinal trajectories. These benefits increase with the CAV penetration rate. Figure 11a,b show the spatial distribution of the average speed when V/C is 0.5 and 1.25, respectively. When all vehicles are HVs and the traffic is undersaturated, the spatial average speed gradually increases in the adjusted control zone within the first 300 m and then begins to slow until it reaches the stop bar (530 m), as depicted in Figure 11a. This is because some vehicles stop and queue up between 300 and 530 m during red lights. When compared, as shown in Figure 11b, due to traffic saturation, the spatial average speed begins to decrease at 200 m. The speed of oversaturated traffic flow at the stop bar is significantly smaller than that of undersaturated traffic flow. As the rate of CAV penetration rises, the speed fluctuations in the control adjustment zone decrease, especially when CAV penetration rates reach 60% or higher under saturated and oversaturated traffic demand. When all vehicles are CAVs, the spatial average speed of the entire control adjustment zone is comparatively reliable due to the smooth transverse trajectories of vehicles provided by trajectory design. The average speed of vehicles passing through the stop bar also increases with increasing CAV penetration rate as a result of the absence of stop-and-go traffic.

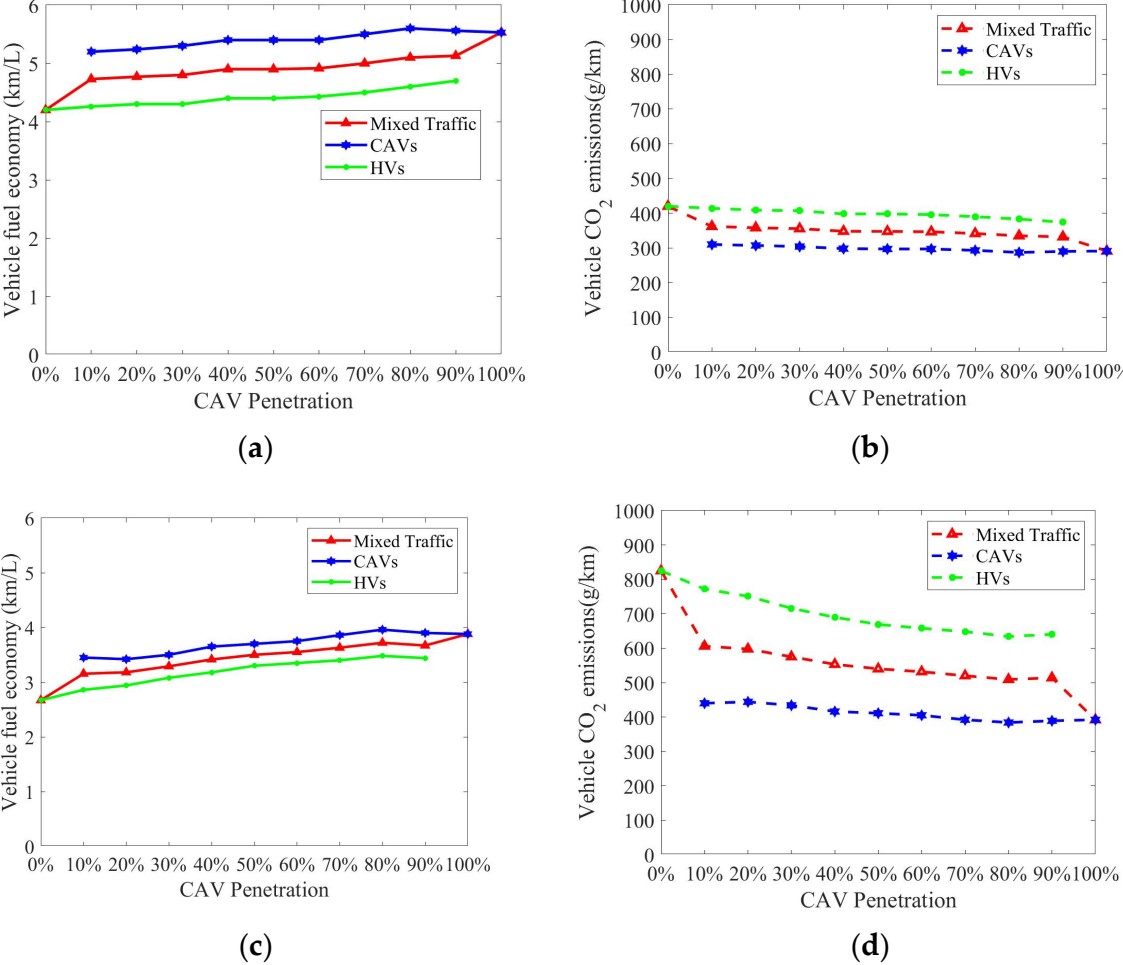

**Figure 10.** *Cont.*

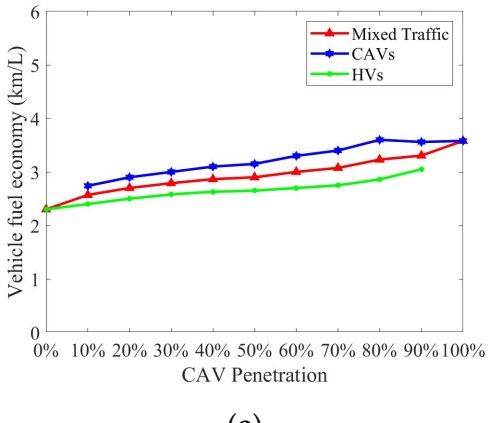

(**e**)

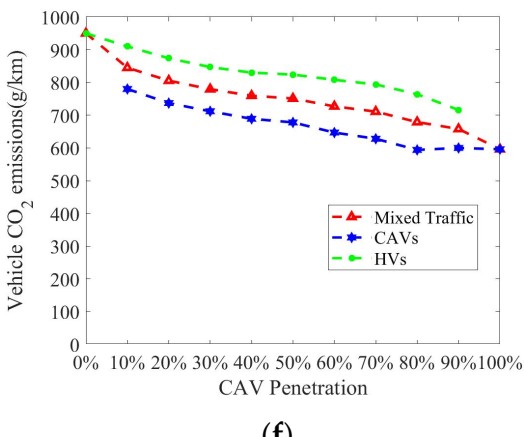

(**f**)

**Figure 10.** Impact of different traffic demands and CAV penetration rates on average fuel economy and $CO_2$ emissions of vehicles. (**a**) Average fuel economy (V/C = 0.50); (**b**) $CO_2$ emissions (V/C = 0.50); (**c**) Average fuel economy (V/C = 1.00); (**d**) $CO_2$ emissions (V/C = 1.00); (**e**) Average fuel economy (V/C = 1.25); (**f**) $CO_2$ emissions (V/C = 1.25).

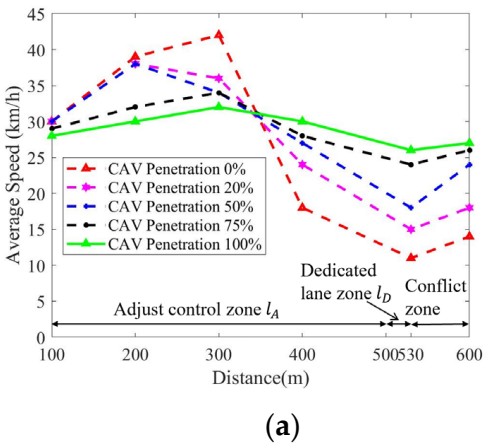

(**a**)

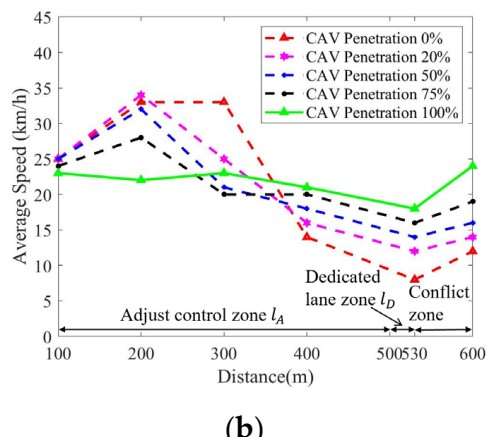

(**b**)

**Figure 11.** Spatial distribution of the spatial average velocity under different CAV penetration rates. (**a**) V/C = 0.50; (**b**) V/C = 1.25.

Figure 12a–c present the average number of lane changes for heterogeneous traffic flow, CAVs, and HVs under different traffic demand conditions for various CAV penetration rates. Although the number of lane changes varies with the demand and penetration rate of CAVs, the overall difference is not important. The typical number of lane changes for CAVs is higher than that for HVs, and the lane change situation is less affected by traffic demand, indicating a higher robustness of lane changing. There are two reasons for this: on the one hand, the setting of pre-signal lights allows HVs to have at most one lane change opportunity to avoid invalid lane changing behavior and reduce traffic conflicts in the control adjustment zone. In addition, to fully utilize the spatiotemporal resources near the intersection, CAVs avoid stopping at the stop line and slow within the control adjustment zone, forming a new queue.

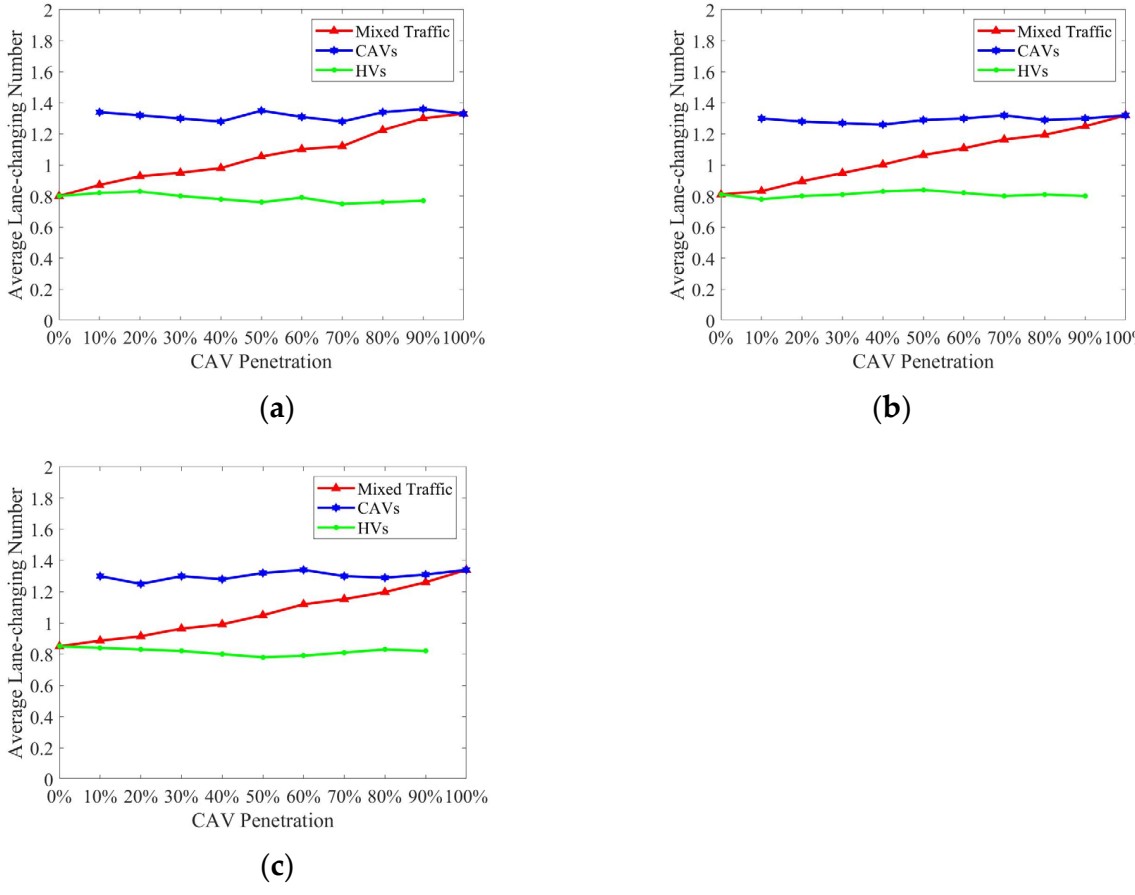

**Figure 12.** Impacts on the number of CAVs that change lanes under various demand levels. (**a**) Average lane-changing number (V/C = 0.50); (**b**) Average lane-changing number (V/C = 1.00); (**c**) Average lane-changing number (V/C = 1.25).

## 5. Conclusions and Recommendation

This article proposes a new traffic management method for heterogeneous traffic flow signal control and CAV trajectory optimization established based on pre-signal lights and dedicated CAV lanes. The proposed method integrates conventional signal timing strategies, dedicated CAV lanes, and CAV trajectory planning close to the lane approach. The method is used to manage mixed traffic of CAVs and HVs. Built with discrete time, the control system constructs a two-layer optimization model for intersection signal phase duration and CAV trajectory planning built on the collection of HV light semantic information, CAV turning demand information, and position and velocity information of all vehicles in the control adjustment zone and dedicated lane zone. The upper-layer model optimizes the phase duration in real time based on the actual total number and type of vehicles entering the control adjustment zone, and the lower-layer model optimizes the CAV lane change strategy and vehicle acceleration optimization curve based on the phase duration optimized by the upper-layer model. Based on the Webster optimal cycle formula for calculating the optimal phase duration, an improved cuckoo search algorithm combined with the EO algorithm is designed to solve the model efficiently. Numerical studies verify the advantages of the proposed method based on pre-signal lights and dedicated CAV lanes for heterogeneous traffic flow signal control and CAV trajectory optimization. The capacity of the intersection is significantly improved and $CO_2$ emissions and the average fuel consumption and lane change frequency of CAVs can be significantly reduced, while the traffic speed and delay of heterogeneous traffic flow are significantly improved.

In this study, the signal timing plan for the intersection is built on the Webster optimal cycle formula, which can be improved. The spatiotemporal resources of the intersection

conflict area with dedicated CAV lanes will be explored in our future research. Another challenge is to consider conventional vehicles that are both unobservable and uncontrollable without pre-signal lights or request for information zones. Estimating and predicting the states of conventional vehicles is another difficulty. In addition, from the intersection level to the road network level, we intend to expand the CAV trajectory planning model.

**Author Contributions:** Conceptualization, J.W. and H.Y.; methodology, J.W.; software, S.C.; validation, J.W., Z.Y. and S.C.; formal analysis, H.Y.; investigation, H.Y.; resources, Z.Y.; data curation, J.W.; writing—original draft preparation, S.C.; writing—review and editing, Y.R.; visualization, S.C.; supervision, Y.R.; project administration, Y.R.; funding acquisition, Y.R. All authors have read and agreed to the published version of the manuscript.

**Funding:** This research was funded by the Beijing Municipal Science & Technology Plan (Grant No. Z211100004221008), the National Natural Science Foundation of China (Grant No. 51878020) and the National Natural Science Foundation of China (Grant No. 51908018).

**Institutional Review Board Statement:** Not applicable.

**Informed Consent Statement:** Not applicable.

**Data Availability Statement:** Not applicable.

**Conflicts of Interest:** The authors declare no conflict of interest.

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
