# Peer review of "Heterogeneous Traffic Flow Signal Control and CAV Trajectory Optimization Based on Pre-Signal Lights and Dedicated CAV Lanes"

_sustainability, doi:10.3390/su152115295_

Round 1

Reviewer 1 Report

Dear Authors;

The manuscript titled “Heterogeneous Traffic Flow Signal Control and CAV Trajectory Optimization Based on Pre-signal Lights and Dedicated” is evaluated meticulously from an academic perspective. 

Kind regards,  

Author Response

Please review the submitted documentation.

Reviewer 2 Report

In this paper, the authors studied heterogeneous traffic flow signal control and CAV trajectory optimization. However, I have some comments.

1.      Please perform thorough revision to remove any grammatical errors. Some references are also missing.

2.      What are the key objectives and benefits of utilizing dedicated connected and automated vehicle (CAV) lanes in the context of traffic system efficiency and vehicle emissions reduction?

3.       How do researchers typically approach the integration of cellular to vehicle (C2V) technology in studies related to CAVs, and what is the significance of considering lane-changing behavior in CAVs?

4.      What challenges arise when attempting to manage traffic in a mixed operation of human-operated vehicles (HVs) and CAVs, and how does the suggested traffic management technique address these challenges?

5.      Can you elaborate on the two-layer optimization model used for signal duration calculation and CAV trajectory planning? What types of information are collected and utilized for optimization in this model?

6.      How does the study propose to measure the success of the suggested signal control and CAV trajectory optimization method? Are there specific metrics or data points that were used to demonstrate its benefits?

7.      Could you provide more details on the improved cuckoo algorithm mentioned in the text? How does it contribute to solving the optimization model, and what makes it effective in this context?

8.      Literature review is incomplete. Latest works in technical literature which study sustainable resource allocation in futuristic networks should be added such as:

a. Ranjha, A., Javed, M.A., Srivastava, G. and Lin, J.C.W., 2022. Intercell Interference Coordination for UAV enabled URLLC with perfect/imperfect CSI using cognitive radio. IEEE Open Journal of the Communications Society. 

b. Hussien, M., Abdelmoaty, A., Elsaadany, M., Ahmed, M.F., Gagnon, G., Nguyen, K.K. and Cheriet, M., 2023. Carrier frequency offset estimation in 5G NR: Introducing gradient boosting machines. IEEE Access, 11, pp.34128-34137.

c. Ranjha, A., Javed, M.A., Srivastava, G. and Asif, M., 2023. Quasi-Optimization of Resource Allocation and Positioning for Solar-Powered UAVs. IEEE Transactions on Network Science and Engineering. 

Typos should be corrected. Some references are missing, they should be added.

Author Response

(The authors gave the same response as above.)

Reviewer 3 Report

This paper will accept if the following amendment done by the author:

·       The abstract is wordy and not informative. The structure of the abstract needs revision.

·       English language needs some improvement throughout the paper. Example in the Abstract “Moreover, which found that present results found to be the present results”

·       On what basis the parametric values are chosen? Do they correspond to a specific physical condition? Please justify?

·       Authors should related to their work with real life application which specified?

·       Improve the results and discussion section with physically?

·       Author check the equations carefully and also give the reference of governing model?

·       The literature review suffers from significant self-citations and is not comprehensive given the many effects and should be improved by considering the related references:

https://doi.org/10.1177/1687814020968322

https://link.springer.com/article/10.1007/s10973-021-10913-0

https://doi.org/10.3390/nano12071207

·       The introduction is very poorly written like as ‘‘check the grammar?

All the comments are provided in the review comments. 

Author Response

(The authors gave the same response as above.)

Round 2

Reviewer 2 Report

all comments have been addressed